# Assessing Cognitive Factors of Modular Distance Learning of K-12 Students Amidst the COVID-19 Pandemic towards Academic Achievements and Satisfaction

**DOI:** 10.3390/bs12070200

**Published:** 2022-06-21

**Authors:** Yung-Tsan Jou, Klint Allen Mariñas, Charmine Sheena Saflor

**Affiliations:** 1Department of Industrial and Systems Engineering, Chung Yuan Christian University, Taoyuan 320, Taiwan; ytjou@cycu.edu.tw (Y.-T.J.); saflorcharmine@yahoo.com (C.S.S.); 2School of Industrial Engineering and Engineering Management, Mapua University, Manila 1002, Philippines; 3Department of Industrial Engineering, Occidental Mindoro State College, San Jose 5100, Philippines

**Keywords:** COVID-19, K-12, modular distance learning, cognitive capacity, SEM

## Abstract

The COVID-19 pandemic brought extraordinary challenges to K-12 students in using modular distance learning. According to Transactional Distance Theory (TDT), which is defined as understanding the effects of distance learning in the cognitive domain, the current study constructs a theoretical framework to measure student satisfaction and Bloom’s Taxonomy Theory (BTT) to measure students’ academic achievements. This study aims to evaluate and identify the possible cognitive capacity influencing K-12 students’ academic achievements and satisfaction with modular distance learning during this new phenomenon. A survey questionnaire was completed through an online form by 252 K-12 students from the different institutions of Occidental Mindoro. Using Structural Equation Modeling (SEM), the researcher analyses the relationship between the dependent and independent variables. The model used in this research illustrates cognitive factors associated with adopting modular distance learning based on students’ academic achievements and satisfaction. The study revealed that students’ background, experience, behavior, and instructor interaction positively affected their satisfaction. While the effects of the students’ performance, understanding, and perceived effectiveness were wholly aligned with their academic achievements. The findings of the model with solid support of the integrative association between TDT and BTT theories could guide decision-makers in institutions to implement, evaluate, and utilize modular distance learning in their education systems.

## 1. Introduction

The 2019 coronavirus is the latest infectious disease to develop rapidly worldwide [1], affecting economic stability, global health, and education. Most countries have suspended thee-to-face classes in order to curb the spread of the virus and reduce infections [2]. One of the sectors impacted has been education, resulting in the suspension of face-to-face classes to avoid spreading the virus. The Department of Education (DepEd) has introduced modular distance learning for K-12 students to ensure continuity of learning during the COVID-19 pandemic. According to Malipot (2020), modular learning is one of the most popular sorts of distance learning alternatives to traditional face-to-face learning [3]. As per DepEd’s Learner Enrolment and Survey Forms, 7.2 million enrollees preferred “modular” remote learning, TV and radio-based practice, and other modalities, while two million enrollees preferred online learning. It is a method of learning that is currently being used based on the preferred distance learning mode of the students and parents through the survey conducted by the Department of Education (DepEd); this learning method is mainly done through the use of printed and digital modules [4]. It also concerns first-year students in rural areas; the place net is no longer available for online learning. Supporting the findings of Ambayon (2020), modular teaching within the teach-learn method is more practical than traditional educational methods because students learn at their own pace during this modular approach. This educational platform allows K-12 students to interact in self-paced textual matter or digital copy modules. With these COVID-19 outbreaks, some issues concerned students’ academic, and the factors associated with students’ psychological status during the COVID-19 lockdown [5].

Additionally, this new learning platform, modular distance learning, seems to have impacted students’ ability to discover and challenged their learning skills. Scholars have also paid close attention to learner satisfaction and academic achievement when it involves distance learning studies and have used a spread of theoretical frameworks to assess learner satisfaction and educational outcomes [6,7]. Because this study aimed to boost academic achievement and satisfaction in K-12 students, the researcher thoroughly applied transactional distance theory (TDT) to understand the consequences of distance in relationships in education. The TDT was utilized since it has the capability to establish the psychological and communication factors between the learners and the instructors in distance education that could eventually help researchers in identifying the variables that might affect students’ academic achievement and satisfaction [8]. In this view, distance learning is primarily determined by the number of dialogues between student and teacher and the degree of structuring of the course design. It contributes to the core objective of the degree to boost students’ modular learning experiences in terms of satisfaction. On the other hand, Bloom’s Taxonomy Theory (BTT) was applied to investigate the students’ academic achievements through modular distance learning [6]. Bloom’s theory was employed in addition to TDT during this study to enhance students’ modular educational experiences. Moreover, TDT was utilized to check students’ modular learning experiences in conjuction with enhacing students’ achievements.

This study aimed to detect the impact of modular distance learning on K-12 students during the COVID-19 pandemic and assess the cognitive factors affecting academic achievement and student satisfaction. Despite the challenging status of the COVID-19 outbreak, the researcher anticipated a relevant result of modular distance learning and pedagogical changes in students, including the cognitive factors identified during this paper as latent variables as possible predictors for the utilization of K-12 student academic achievements and satisfaction.

### 1.1. Theoretical Research Framework 

This study used TDT to assess student satisfaction and Bloom’s theory to quantify academic achievement. It aimed to assess the impact of modular distance learning on academic achievement and student satisfaction among K-12 students. The Transactional Distance Theory (TDT) was selected for this study since it refers to student-instructor distance learning. TDT Moore (1993) states that distance education is “the universe of teacher-learner connections when learners and teachers are separated by place and time.” Moore’s (1990) concept of ”Transactional Distance” adopts the distance that occurs in all linkages in education, according to TDT Moore (1993). Transactional distance theory is theoretically critical because it states that the most important distance is transactional in distance education, rather than geographical or temporal [9,10]. According to Garrison (2000), transactional distance theory is essential in directing the complicated experience of a cognitive process such as distance teaching and learning. TDT evaluates the role of each of these factors (student perception, discourse, and class organization), which can help with student satisfaction research [11]. Bloom’s Taxonomy is a theoretical framework for learning created by Benjamin Bloom that distinguishes three learning domains: Cognitive domain skills center on knowledge, comprehension, and critical thinking on a particular subject. Bloom recognized three components of educational activities: cognitive knowledge (or mental abilities), affective attitude (or emotions), and psychomotor skills (or physical skills), all of which can be used to assess K-12 students’ academic achievement. According to Jung (2001), “Transactional distance theory provides a significant conceptual framework for defining and comprehending distance education in general and a source of research hypotheses in particular,” shown in Figure 1 [12].

### 1.2. Hypothesis Developments and Literature Review

This section will discuss the study hypothesis and relate each hypothesis to its related studies from the literature.

**Hypothesis** **1.***There is a significant relationship between students’ background and students’ behavior*. 

The teacher’s guidance is essential for students’ preparedness and readiness to adapt to a new educational environment. Most students opt for the Department of Education’s “modular” distance learning options [3]. Analyzing students’ study time is critical for behavioral engagement because it establishes if academic performance is the product of student choice or historical factors [13].

**Hypothesis** **2.***There is a significant relationship between students’ background and students’ experience*.

Modules provide goals, experiences, and educational activities that assist students in gaining self-sufficiency at their speed. It also boosts brain activity, encourages motivation, consolidates self-satisfaction, and enables students to remember what they have learned [14]. Despite its success, many families face difficulties due to their parents’ lack of skills and time [15].

**Hypothesis** **3.***There is a significant relationship between students’ behavior and students’ instructor interaction*. 

Students’ capacity to answer problems reflects their overall information awareness [5]. Learning outcomes can either cause or result in students and instructors behavior. Students’ reading issues are due to the success of online courses [16].

**Hypothesis** **4.***There is a significant relationship between students’ experience and students’ instructor interaction*.

The words “student experience” relate to classroom participation. They establish a connection between students and their school, teachers, classmates, curriculum, and teaching methods [17]. The three types of student engagement are behavioral, emotional, and cognitive. Behavioral engagement refers to a student’s enthusiasm for academic and extracurricular activities. On the other hand, emotional participation is linked to how children react to their peers, teachers, and school. Motivational engagement refers to a learner’s desire to learn new abilities [18].

**Hypothesis** **5.***There is a significant relationship between students’ behavior and students’ understanding*.

Individualized learning connections, outstanding training, and learning culture are all priorities at the Institute [19,20]. The modular technique of online learning offers additional flexibility. The use of modules allows students to investigate alternatives to the professor’s session [21].

**Hypothesis** **6.***There is a significant relationship between students’ experience and students’ performance*.

Student conduct is also vital in academic accomplishment since it may affect a student’s capacity to study as well as the learning environment for other students. Students are self-assured because they understand what is expected [22]. They are more aware of their actions and take greater responsibility for their learning.

**Hypothesis** **7.***There is a significant relationship between students’ instructor interaction and students’ understanding*.

Modular learning benefits students by enabling them to absorb and study material independently and on different courses. Students are more likely to give favorable reviews to courses and instructors if they believe their professors communicated effectively and facilitated or supported their learning [23].

**Hypothesis** **8.**
*There is a significant relationship between students’ instructor interaction and students’ performance.*


Students are more engaged and active in their studies when they feel in command and protected in the classroom. Teachers play an essential role in influencing student academic motivation, school commitment, and disengagement. In studies on K-12 education, teacher-student relationships have been identified [24]. Positive teacher-student connections improve both teacher attitudes and academic performance. 

**Hypothesis** **9.***There is a significant relationship between students’ understanding and students’ satisfaction*.

Instructors must create well-structured courses, regularly present in their classes, and encourage student participation. When learning objectives are completed, students better understand the course’s success and learning expectations. “Constructing meaning from verbal, written, and graphic signals by interpreting, exemplifying, classifying, summarizing, inferring, comparing, and explaining” is how understanding is characterized [25]. 

**Hypothesis** **10.***There is a significant relationship between students’ performance and student’s academic achievement*.

Academic emotions are linked to students’ performance, academic success, personality, and classroom background [26]. Understanding the elements that may influence student performance has long been a goal for educational institutions, students, and teachers.

**Hypothesis** **11.***There is a significant relationship between students’ understanding and students’ academic achievement*.

Modular education views each student as an individual with distinct abilities and interests. To provide an excellent education, a teacher must adapt and individualize the educational curriculum for each student. Individual learning may aid in developing a variety of exceptional and self-reliant attributes [27]. Academic achievement is the current level of learning in the Philippines [28]. 

**Hypothesis** **12.***There is a significant relationship between students’ performance and students’ satisfaction*.

Academic success is defined as a student’s intellectual development, including formative and summative assessment data, coursework, teacher observations, student interaction, and time on a task [29]. Students were happier with course technology, the promptness with which content was shared with the teacher, and their overall wellbeing [30].

**Hypothesis** **13.***There is a significant relationship between students’ academic achievement and students’ perceived effectiveness*.

Student satisfaction is a short-term mindset based on assessing students’ educational experiences [29]. The link between student satisfaction and academic achievement is crucial in today’s higher education: we discovered that student satisfaction with course technical components was linked to a higher relative performance level [31].

**Hypothesis** **14.**
*There is a significant relationship between students’ satisfaction and students’ perceived effectiveness.*


There is a strong link between student satisfaction and their overall perception of learning. A satisfied student is a direct effect of a positive learning experience. Perceived learning results had a favorable impact on student satisfaction in the classroom [32].

## 2. Materials and Methods

### 2.1. Participants

The principal area under study was San Jose, Occidental Mindoro, although other locations were also accepted. The survey took place between February and March 2022, with the target population of K-12 students in Junior and Senior High Schools from grades 7 to 12, aged 12 to 20, who are now implementing the Modular Approach in their studies during the COVID-19 pandemic. A 45-item questionnaire was created and circulated online to collect the information. A total of 300 online surveys was sent out and 252 online forms were received, a total of 84% response rate [33]. According to several experts, the sample size for Structural Equation Modeling (SEM) should be between 200 and 500 [34].

### 2.2. Questionnaire

The theoretical framework developed a self-administered test. The researcher created the questionnaire to examine and discover the probable cognitive capacity influencing K-12 students’ academic achievement in different parts of Occidental Mindoro during this pandemic as well as their satisfaction with modular distance learning. The questionnaire was designed through Google drive as people’s interactions are limited due to the effect of the COVID-19 pandemic. The questionnaire’s link was sent via email, Facebook, and other popular social media platforms.

The respondents had to complete two sections of the questionnaire. The first is their demographic information, including their age, gender, and grade level. The second is about their perceptions of modular learning. The questionnaire is divided into 12 variables: (1) Student’s Background, (2) Student’s Experience, (3) Student’s Behavior, (4) Student’s Instructor Interaction, (5) Student’s Performance, (6) Student’s Understanding, (7) Student’s Satisfaction, (8) Student’s Academic Achievement, and (9) Student’s Perceived Effectiveness. A 5-point Likert scale was used to assess all latent components contained in the SEM shown in Table 1.

### 2.3. Structural Equation Modeling (SEM)

All the variables have been adapted from a variety of research in the literature. The observable factors were scored on a Likert scale of 1–5, with one indicating “strongly disagree” and five indicating “strongly agree”, and the data were analyzed using AMOS software. Theoretical model data were confirmed by Structural Equation Modeling (SEM). SEM is more suitable for testing the hypothesis than other methods [53]. There are many fit indices in the literature, of which the most commonly used are: CMIN/DF, Comparative Fit Index (CFI), AGFI, GFI, and Root Mean Square Error (RMSEA). Table 2 demonstrates the Good Fit Values and Acceptable Fit Values of the fit indices, respectively. AGFI and GFI are based on residuals; when sample size increases, the value of the AGFI also increase. It takes a value between 0 and 1. The fit is good if the value is more significant than 0.80. GFI is a model index that spans from 0 to 1, with values above 0.80 deemed acceptable. An RMSEA of 0.08 or less suggests a good fit [54], and a value of 0.05 to 0.08 indicates an adequate fit [55]. 

## 3. Results and Discussion 

Figure 2 demonstrates the initial SEM for the cognitive factors of Modular Distance learning towards academic achievements and satisfaction of K-12 students during the COVID-19 pandemic. According to the figure below, three hypotheses were not significant: Students’ Behavior to Students’ Instructor Interaction (Hypothesis 3), Students’ Understanding of Students’ Academic Achievement (Hypothesis 11), and Students’ Performance to Students’ Satisfaction (Hypothesis 12). Therefore, a revised SEM was derived by removing this hypothesis in Figure 3. We modified some indices to enhance the model fit based on previous studies using the SEM approach [47]. Figure 3 demonstrates the final SEM for evaluating cognitive factors affecting academic achievements and satisfaction and the perceived effectiveness of K-12 students’ response to Modular Learning during COVID-19, shown in Table 3. Moreover, Table 4 demonstrates the descriptive statistical results of each indicator. 

The current study was improved by Moore’s transactional distance theory (TDT) and Bloom’s taxonomy theory (BTT) to evaluate cognitive factors affecting academic achievements and satisfaction and the perceived effectiveness of K-12 students’ response toward modular learning during COVID-19. SEM was utilized to analyze the correlation between Student Background (SB), Student Experience (SE), Student Behavior (SBE), Student Instructor Interaction (SI), Student Performance (SP), Student Understanding (SAU), Student Satisfaction (SS), Student’s Academic achievement (SAA), and Student’s Perceived effectiveness (SPE). A total of 252 data samples were acquired through an online questionnaire. 

According to the findings of the SEM, the students’ background in modular learning had a favorable and significant direct effect on SE (β: 0.848, *p* = 0.009). K-12 students should have a background and knowledge in modular systems to better experience this new education platform. Putting the students through such an experience would support them in overcoming all difficulties that arise due to the limitations of the modular platforms. Furthermore, SEM revealed that SE had a significant adverse impact on SI (β: 0.843, *p* = 0.009). The study shows that students who had previous experience with modular education had more positive perceptions of modular platforms. Additionally, students’ experience with modular distance learning offers various benefits to them and their instructors to enhance students’ learning experiences, particularly for isolated learners.

Regarding the Students’ Interaction—Instructor, it positively impacts SAU (β: 0.873, *p* = 0.007). Communication helps students experience positive emotions such as comfort, satisfaction, and excitement, which aim to enhance their understanding and help them attain their educational goals [62]. The results revealed that SP substantially impacted SI (β: 0.765; *p* = 0.005). A student becomes more academically motivated and engaged by creating and maintaining strong teacher-student connections, which leads to successful academic performance.

Regarding the Students’ Understanding Response, the results revealed that SAA (β: 0.307; *p* = 0.052) and SS (β: 0.699; *p* = 0.008) had a substantial impact on SAU. Modular teaching is concerned with each student as an individual and with their specific capability and interest to assist each K-12 student in learning and provide quality education by allowing individuality to each learner. According to the Department of Education, academic achievement is the new level for student learning [63]. Meanwhile, SAA was significantly affected by the Students’ Performance Response (β: 0.754; *p* = 0.014). It implies that a positive performance can give positive results in student’s academic achievement, and that a negative performance can also give negative results [64]. Pekrun et al. (2010) discovered that students’ academic emotions are linked to their performance, academic achievement, personality, and classroom circumstances [26].

Results showed that students’ academic achievement significantly positively affects SPE (β: 0.237; *p* = 0.024). Prior knowledge has had an indirect effect on academic accomplishment. It influences the amount and type of current learning system where students must obtain a high degree of mastery [65]. According to the student’s opinion, modular distance learning is an alternative solution for providing adequate education for all learners and at all levels in the current scenario under the new education policy [66]. However, the SEM revealed that SS significantly affected SPE (β: 0.868; *p* = 0.009). Students’ perceptions of learning and satisfaction, when combined, can provide a better knowledge of learning achievement [44]. Students’ perceptions of learning outcomes are an excellent predictor of student satisfaction.

Since *p*-values and the indicators in Students’ Behavior are below 0.5, therefore two paths connecting SBE to students’ interaction—instructor (0.155) and students’ understanding (0.212) are not significant; thus, the latent variable Students’ Behavior has no effect on the latent variable Students’ Satisfaction and academic achievement as well as perceived effectiveness on modular distance learning of K12 students. This result is supported by Samsen-Bronsveld et al. (2022), who revealed that the environment has no direct influence on the student’s satisfaction, behavior engagement, and motivation to study [67]. On the other hand, the results also showed no significant relationship between Students’ Performance and Students’ Satisfaction (0.602) because the correlation *p*-values are greater than 0.5. Interestingly, this result opposed the other related studies. According to Bossman & Agyei (2022), satisfaction significantly affects performance or learning outcomes [68]. In addition, it was discovered that the main drivers of the students’ performance are the students’ satisfaction [64,69]. 

The result of the study implies that the students’ satisfaction serves as the mediator between the students’ performance and the student-instructor interaction in modular distance learning for K-12 students [70]. 

Table 5 The reliabilities of the scales used, i.e., Cronbach’s alphas, ranged from 0.568 to 0.745, which were in line with those found in other studies [71]. As presented in Table 6, the IFI, TLI, and CFI values were greater than the suggested cutoff of 0.80, indicating that the specified model’s hypothesized construct accurately represented the observed data. In addition, the GFI and AGFI values were 0.828 and 0.801, respectively, indicating that the model was also good. The RMSEA value was 0.074, lower than the recommended value. Finally, the direct, indirect, and total effects are presented in Table 7.

Table 6 shows that the five parameters, namely the Incremental Fit Index, Tucker Lewis Index, the Comparative Fit Index, Goodness of Fit Index, and Adjusted Goodness Fit Index, are all acceptable with parameter estimates greater than 0.8, whereas mean square error is excellent with parameter estimates less than 0.08.

## 4. Conclusions

The education system has been affected by the 2019 coronavirus disease; face-to-face classes are suspended to control and reduce the spread of the virus and infections [2]. The suspension of face-to-face classes results in the application of modular distance learning for K-12 students according to continuity of learning during the COVID-19 pandemic. With the outbreak of COVID-19, some issues concerning students’ academic Performance and factors associated with students’ psychological status are starting to emerge, which impacted the students’ ability to learn. This study aimed to perceive the impact of Modular Distance learning on the K-12 students amid the COVID-19 pandemic and assess cognitive factors affecting students’ academic achievement and satisfaction.

This study applied Transactional Distance Theory (TDT) and Bloom Taxonomy Theory (BTT) to evaluate cognitive factors affecting students’ academic achievements and satisfaction and evaluate the perceived effectiveness of K-12 students in response to modular learning. This study applied Structural Equation Modeling (SEM) to test hypotheses. The application of SEM analyzed the correlation among students’ background, experience, behavior, instructor interaction, performance, understanding, satisfaction, academic achievement, and student perceived effectiveness.

A total of 252 data samples were gathered through an online questionnaire. Based on findings, this study concludes that students’ background in modular distance learning affects their behavior and experience. Students’ experiences had significant effects on the performance and understanding of students in modular distance learning. Student instructor interaction had a substantial impact on performance and learning; it explains how vital interaction with the instructor is. The student interacting with the instructor shows that the student may receive feedback and guidance from the instructor. Understanding has a significant influence on students’ satisfaction and academic achievement. Student performance has a substantial impact on students’ academic achievement and satisfaction. Perceived effectiveness was significantly influenced by students’ academic achievement and student satisfaction. However, students’ behavior had no considerable effect on students’ instructor interaction, and students’ understanding while student performance equally had no significant impact on student satisfaction. From this study, students are likely to manifest good performance, behavior, and cognition when they have prior knowledge with regard to modular distance learning. This study will help the government, teachers, and students take the necessary steps to improve and enhance modular distance learning that will benefit students for effective learning.

The modular learning system has been in place since its inception. One of its founding metaphoric pillars is student satisfaction with modular learning. The organization demonstrated its dedication to the student’s voice as a component of understanding effective teaching and learning. Student satisfaction research has been transformed by modular learning. It has caused the education research community to rethink long-held assumptions that learning occurs primarily within a metaphorical container known as a “course.” When reviewing studies on student satisfaction from a factor analytic perspective, one thing becomes clear: this is a complex system with little consensus. Even the most recent factor analytical studies have done little to address the lack of understanding of the dimensions underlying satisfaction with modular learning. Items about student satisfaction with modular distance learning correspond to forming a psychological contract in factor analytic studies. The survey responses are reconfigured into a smaller number of latent (non-observable) dimensions that the students never really articulate but are fully expected to satisfy. Of course, instructors have contracts with their students. Studies such as this one identify the student’s psychological contact after the fact, rather than before the class. The most important aspect is the rapid adoption of this teaching and learning mode in Senior High School. Another balancing factor is the growing sense of student agency in the educational process. Students can express their opinions about their educational experiences in formats ranging from end-of-course evaluation protocols to various social networks, making their voices more critical.

Furthermore, they all agreed with latent trait theory, which holds that the critical dimensions that students differentiate when expressing their opinions about modular learning are formed by the combination of the original items that cannot be directly observed—which underpins student satisfaction. As stated in the literature, the relationship between student satisfaction and the characteristic of a psychological contract is illustrated. Each element is translated into how it might be expressed in the student’s voice, and then a contract feature and an assessment strategy are added. The most significant contributor to the factor pattern, engaged learning, indicates that students expect instructors to play a facilitative role in their teaching. This dimension corresponds to the relational contract, in which the learning environment is stable and well organized, with a clear path to success.

## 5. Limitations and Future Work

This study was focused on the cognitive capacity of modular distance learning towards academic achievements and satisfaction of K-12 students during the COVID-19 pandemic. The sample size in this study was small, at only 252. If this study is repeated with a larger sample size, it will improve the results. The study’s restriction was to the province of Occidental Mindoro; Structural Equation Modeling (SEM) was used to measure all the variables. Thus, this will give an adequate solution to the problem in the study.

The current study underlines that combining TDT and BTT can positively impact the research outcome. The contribution the current study might make to the field of modular distance learning has been discussed and explained. Based on this research model, the nine (9) factors could broadly clarify the students’ adoption of new learning environment platform features. Thus, the current research suggests that more investigation be carried out to examine relationships among the complexity of modular distance learning.

## Figures and Tables

**Figure 1 behavsci-12-00200-f001:**
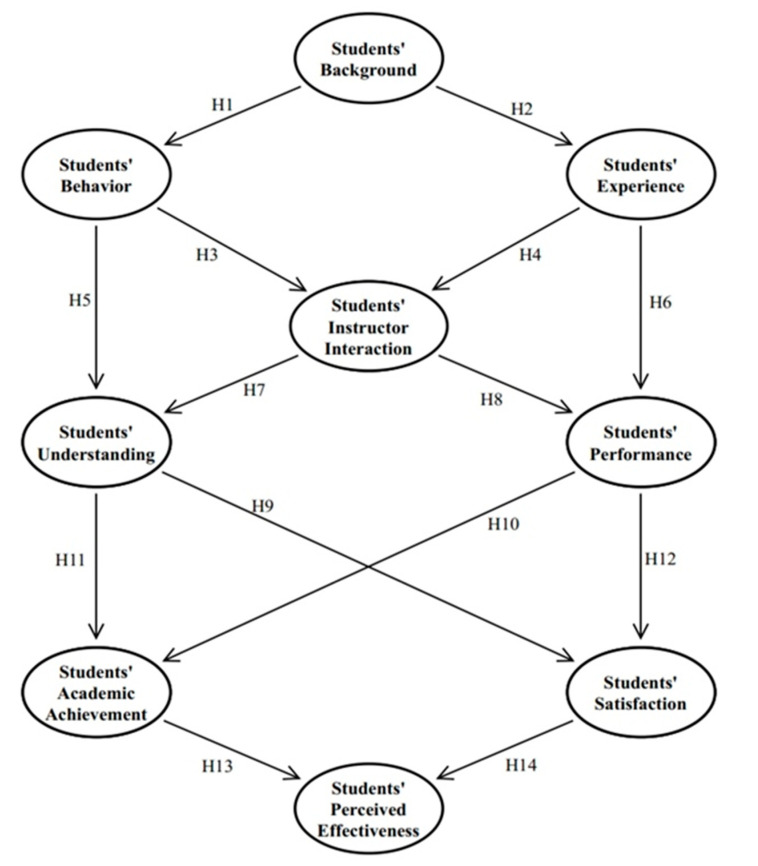
Theoretical Research Framework.

**Figure 2 behavsci-12-00200-f002:**
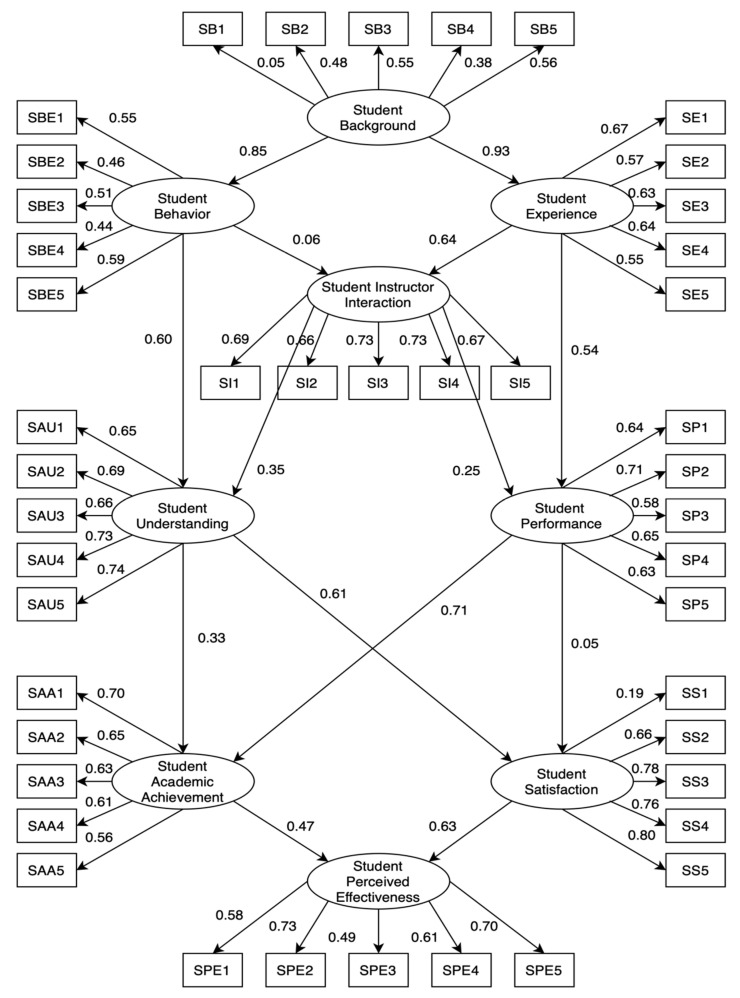
Initial SEM with indicators for evaluating the cognitive factors of modular distance learning towards academic achievements and satisfaction of K-12 students during COVID-19 pandemic.

**Figure 3 behavsci-12-00200-f003:**
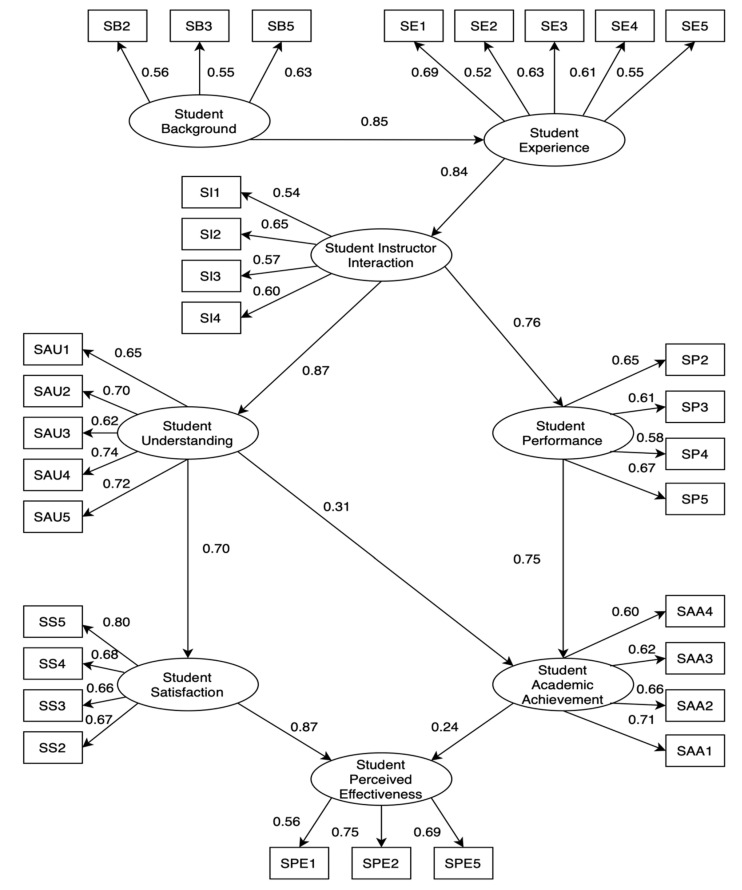
Revised SEM with indicators for evaluating the cognitive factors of modular distance learning towards academic achievements and satisfaction of K-12 students during the COVID-19 pandemic.

**Table 1 behavsci-12-00200-t001:** The construct and measurement items.

Construct	Items	Measures	Supporting Reference
Students’ Background (SB)	SB1	Are you having difficulty with Modular Distance Learning.	Pe Dangle, Y.R. (2020) [35]
	SB2	I prefer Modular Distance Learning Rather than traditional face-to-face training, I prefer the Modular Distance Learning Approach.	Aksan, J.A. (2021) [36]
	SB3	Modular learning aids students in increasing their productivity in education and learning while promoting flexibility in terms of content, time, and space.	Shuja, A. et al. (2019) [37]
	SB4	I have a lot of time to answer the activities with a modular teaching technique.	Aksan, J.A. (2021) [36]
	SB5	I acquire the same amount of learning from using the module as I do from learning in a face-to-face or classroom situation.	Natividad, E. (2021) [38]
Students’ Behavior (SBE)	SBE1	I feel confident in studying and performing well in the modular class.	Delfino, A.P. (2019) [39]
	SBE2	I employed rehearsing techniques like reviewing my notes over and over again.	Lowerison et al. (2006) [40]
	SBE3	I can recall my understanding from the past and help me to understand words.	Santillan, S.C. et al. (2021) [41]
	SBE4	I retain a critical mindset throughout my studies, considering before accepting or rejecting.	Bordeos (2021) [42]
	SBE5	Usually I plan my weekly module work in advance.	Karababa et al. (2010) [43]
Students’ Experience (SE)	SE1	The way the module materials were presented helped to maintain my interest.	Allen et al. (2020) [44]
	SE2	I do not experience any problems during modular distance learning	Amir al (2020) [45]
	SE3	The instructions for completing the assessed tasks were simple to understand.	Santillan et al. (2021) [41]
	SE4	During distance learning, I am not stressed.	Amir et al. (2020) [45]
	SE5	The study workload on this module fitted with my personal circumstances.	Allen et al. (2020) [44]
Students’ Instructor Interaction (SI)	SI1	The instructor updated me on my progress in the course regularly.	Gray & DiLoreto (2020) [46]
	SI2	On this subject, I was satisfied with my teacher’s assistance.	Allen et al. (2020) [44]
	SI3	I kept in touch with the course’s instructor regularly.	Gray & DiLoreto (2020) [46]
	SI4	The instructor was concerned about my performance in this class.	Gray & DiLoreto (2020) [46]
	SI5	My teacher feedback on assessed tasks helped me prepare for the next assessment.	Allen et al. (2020) [44]
Students’ Understanding (SAU)	SAU1	Modular Distance Learning allows me to take my time to understand my school works.	Abuhassna et al. (2020) [6]
	SAU2	The distance learning program met my expectations in terms of quality.	Woolf et al. (2020) [47]
	SAU3	Modular Distance Learning helps me to improve my understanding and skills and also helps to gather new knowledge.	Bordeos (2021) [42]
	SAU4	Modular Distance Learning is a helpful tool to get so focused on activities in my classes.	Abuhassna et al. (2020) [6]
	SAU5	Modular Distance Learning motivates me to study more about the course objectives.	Abuhassna et al. (2020) [6]
Students’ Performance (SP)	SP1	I can effectively manage my study time and complete assignments on schedule.	Richardson and Swan (2003) [48]
	SP2	When completing projects or participating in class discussions, combine ideas or concepts from several courses.	Delfino, A.P. (2019) [39]
	SP3	I employed elaboration techniques like summarizing the material and relating it to previous knowledge.	Lowerison et al. (2006) [40]
	SP4	In my studies, I am self-disciplined and find it easy to schedule reading and homework time.	Richardson and Swan (2003) [48]
	SP5	I was confident in my capacity to learn and do well in class.	Delfino, A.P. (2019) [39]
Student’s Academic Achievement (SAA)	SAA1	I have more opportunities to reflect on what I’ve learned in modular classes.	Dziuban et al. (2015) [49]
	SAA2	I am committed to completing my homework (readings, assignments) on time and engaging fully in class discussions.	Mt. San Antonio College (2012) [50]
	SAA3	My modular learning experience has increased my opportunity to access and use information.	Dziuban et al. (2015) [49]
	SAA4	I employed assessment, evaluation, and criticizing procedures for assessing, evaluating, and critiquing the material.	Lowerison et al. (2006) [40]
	SAA5	I am skilled at juggling many responsibilities while working under time constraints.	Estelami (2013) [51]
Students’ Satisfaction (SS)	SS1	I am always interested in learning about new things.	Abuhassna et al. (2020) [6]
	SS2	I study more efficiently with distance learning.	Amir (2020) [45]
	SS3	Modular learning suits me better than face-to-face classes.	Abuhassna et al. (2020) [6]
	SS4	I prefer distance learning to classroom learning.	Amir et al. (2020) [45]
	SS5	Overall, I am pleased with the module’s quality.	Santillan, S.C. et al. (2021) [41]
Students’ Perceived Effectiveness (SPE)	SPE1	I made use of learning possibilities and resources in this modular distance learning.	Lowerison et al. (2006) [40]
	SPE2	I would recommend modular distance learning study to other students.	Abuhassna et al. (2020) [6]
	SPE3	These classes also challenge me to conduct more independent research and not rely on a single source of information.	Mt. San Antonio College (2012) [50]
	SPE4	Overall, this modular distance learning has been a good platform for studying during the pandemic.	Lowerison et al., 2006) [40]
	SPE5	Overall, I am satisfied with this modular distance learning course.	Aman (2009) [52]

**Table 2 behavsci-12-00200-t002:** Acceptable Fit Values.

Fit Indices	Acceptable Range	Reference
CMIN/DF	<3.00	Norberg et al., 2007 [56]; Li et al., 2013 [57]
GFI	≥0.80	Doloi et al., 2012 [54]
CFI	>0.70	Norberg et al., 2007 [56]; Chen et al., 2012 [58]
RMSEA	≤0.08	Doloi et al., 2012 [54]
AGFI	>0.08	Jaccard and Wan (1996) [59]
TLI	>0.08	Jafari et al., 2021 [60]
IFI	>0.08	Lee et al., 2015 [61]

**Table 3 behavsci-12-00200-t003:** Summary of the Results.

	Hypothesis	*p*-Value	Interpretation
H1	There is a significant relationship between Students’ Background and Students’ Behavior	0.001	Significant
H2	There is a significant relationship between Students’ Background and Students’ Experiences.	0.001	Significant
H3	There is a significant relationship between Students’ Behavior and Students’ instructor Interaction.	0.155	Not Significant
H4	There is a significant relationship between Students’ experience and Students—Interaction	0.020	Significant
H5	There is a significant relationship between Students’ Behavior and Students’ Understanding	0.212	Not Significant
H6	There is a significant relationship between Students’ experience and Students’ Performance	0.001	Significant
H7	There is a significant relationship between Students’ instructor Interaction and Students’ Understanding	0.008	Significant
H8	There is a significant relationship between Students’ Instructor—Interaction and students’ Performance	0.018	Significant
H9	There is a significant relationship between students’ Understanding and Students’ Satisfaction	0.001	Significant
H10	There is a significant relationship between students’ Performance and Students’ Academic Achievement	0.001	Significant
H11	There is a significant relationship between students’ understanding and Students’ Academic Achievement	0.001	Significant
H12	There is a significant relationship between students’ Performance and Students Satisfaction	0.602	Not Significant
H13	There is a significant relationship between Students’ Academic Achievement and students’ Perceived Effectiveness	0.001	Significant
H14	There is a significant relationship between students’ Satisfaction and Students’ Perceived Effectiveness	0.001	Significant

**Table 4 behavsci-12-00200-t004:** Descriptive statistic results.

Factor	Item	Mean	SD	Factor Loading	
				Initial Model	Final Model
Students’ Background	SB1	3.437	0.9147	0.052	-
	SB2	3.000	1.1707	0.480	0.562
	SB3	3.663	0.8042	0.551	0.551
	SB4	3.742	0.8797	0.381	-
	SB5	3.024	1.0746	0.557	0.629
Students’ Behavior	SBE1	3.667	0.8468	0.551	-
	SBE2	3.829	0.8075	0.463	-
	SBE3	3.873	0.7305	0.507	-
	SBE4	3.833	0.8157	0.437	-
	SBE 5	3.849	0.8188	0.585	-
Students’ Experience	SE1	3.853	0.8458	0.669	0.686
	SE2	2.825	1.0643	0.572	0.525
	SE3	3.591	0.9035	0.630	0.634
	SE4	2.758	1.1260	0.639	0.611
	SE5	3.615	0.8693	0.552	0.551
Students’ Instructor Interaction	SI1	3.754	0.7698	0.689	-
	SI2	3.817	0.9095	0.655	0.541
	SI3	3.730	0.9269	0.731	0.645
	SI4	3.929	0.7487	0.685	0.568
	SI5	3.909	0.7805	0.669	0.597
Students’ Understanding	SAU1	4.028	0.8152	0.647	0.652
	SAU2	3.464	0.8390	0.691	0.704
	SAU3	3.873	0.8325	0.658	0.620
	SAU4	3.750	0.8775	0.731	0.741
	SAU5	3.794	0.8591	0.740	0.717
Students’ Performance	SP1	3.762	0.8602	0.640	-
	SP2	3.881	0.7995	0.708	0.655
	SP3	3.778	0.8781	0.582	0.606
	SP4	3.905	0.7927	0.647	0.585
	SP5	3.976	0.8275	0.630	0.673
Student’s Academic Achievement	SAA1	3.885	0.7976	0.696	0.713
	SAA2	3.929	0.7749	0.653	0.658
	SAA3	3.762	0.8319	0.632	0.615
	SAA4	3.837	0.7790	0.612	0.597
	SAA5	3.694	0.8959	0.559	-
Students’ Satisfaction	SS1	4.087	0.7835	0.189	-
	SS2	3.361	0.9943	0.657	0.669
	SS3	2.960	1.1390	0.779	0.659
	SS4	2.889	1.1516	0.759	0.677
	SS5	3.377	1.0918	0.802	0.803
Students’ Perceived Effectiveness	SPE1	3.829	0.7875	0.580	0.558
	SPE2	3.405	1.0153	0.730	0.750
	PE3	3.790	0.8178	0.490	-
	PE4	3.813	0.9367	0.614	-
	PE5	3.492	1.0000	0.696	0.690

**Table 5 behavsci-12-00200-t005:** Construct Validity Model.

Factor	Number of Items	Cronbach’s α
Students’ Background	3	0.598
Students’ Behavior	5	0.682
Students’ Experience	5	0.761
Students’ Instructor Interaction	5	0.817
Students’ Understanding	5	0.825
Students’ Performance	5	0.768
Students’ Academic Achievement	5	0.770
Students’ Satisfaction	5	0.777
Students’ Perceived Effectiveness	5	0.772
Total		0.752

**Table 6 behavsci-12-00200-t006:** Model Fit.

Goodness of Fit Measures of SEM	Parameter Estimates	Minimum Cut-Off	Interpretation
CMIN/DF	2.375	<3.0	Acceptable
Comparative Fit Index (CFI)	0.830	>0.8	Acceptable
Incremental Fit Index (IFI)	0.832	>0.8	Acceptable
Tucker Lewis Index (TLI)	0.812	>0.8	Acceptable
Goodness of Fit Index (GFI)	0.812	>0.8	Acceptable
Adjusted Goodness of Fit Index (AGFI)	0.803	>0.8	Acceptable
Root Mean Square Error (RMSEA)	0.074	<0.08	Acceptable

**Table 7 behavsci-12-00200-t007:** Direct effect, indirect effect, and total effect.

No.	Variable	Direct Effects	*p*-Value	Indirect Effects	*p*-Value	Total Effects	*p*-Value
1	SB–SE	0.848	0.009	-	-	0.848	0.009
2	SB–SI	-	-	0.715	0.006	0.715	0.006
3	SB–SAU	-	-	0.624	0.006	0.624	0.006
4	SB–SP	-	-	0.547	0.004	0.547	0.004
5	SB–SAA	-	-	0.604	0.006	0.604	0.006
6	SB–SS	-	-	0.436	0.006	0.436	0.006
7	SB–SPE	-	-	0.522	0.007	0.522	0.006
8	SE–SI	0.843	0.009	-	-	0.843	0.009
9	SE–SAU	-	-	0.736	0.010	0.736	0.010
10	SE–SP	-	-	0.645	0.005	0.645	0.005
11	SE–SAA	-	-	0.713	0.006	0.713	0.006
12	SE–SS	-	-	0.514	0.006	0.514	0.006
13	SE–SPE	-	-	0.615	0.006	0.615	0.006
14	SI–SAU	0.873	0.007	-	-	0.873	0.007
15	SI–SP	0.765	0.005	-	-	0.765	0.005
16	SI–SAA	-	-	0.845	0.004	0.845	0.004
17	SI–SS	-	-	0.610	0.004	0.610	0.004
18	SI–SPE	-	-	0.730	0.007	0.730	0.007
19	SAU–SP	-	-	-	-	-	-
20	SAU–SAA	0.307	0.052	-	-	0.307	0.052
21	SAU–SS	0.699	0.008	-	-	0.699	0.008
22	SAU–SPE	-	-	0.680	0.011	0.680	0.011
23	SP–SAA	0.754	0.014	-	-	0.754	0.014
24	SP–SS	-	-	-	-	-	-
25	SP–SPE	-	-	0.179	0.018	0.179	0.018
26	SAA–SS	-	-	-	-	-	-
27	SAA–SPE	0.237	0.024	-	-	0.237	0.024
28	SS–SPE	0.868	0.009	-	-	0.868	0.009

## Data Availability

Not applicable.

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
