# Peer review of "Assessing Cognitive Factors of Modular Distance Learning of K-12 Students Amidst the COVID-19 Pandemic towards Academic Achievements and Satisfaction"

_behavsci, 2022, doi:10.3390/bs12070200_

Round 1
Reviewer 1 Report
In this study „A cognitive Capacity of Modular Distance learning towards academic achievements and satisfaction of K-12 students amidst the COVID-19 Pandemic“ aimed to spot the impact of modular distance learning on the K-12 students amid the COVID-19 pandemic and assess the cognitive factors affecting academic achievement and student satisfaction.
The results of this investigation support the hypothesis that mental traits have the most significant, the biggest, and most important impact on student performance.
Theoretical research framework, materials and methods is described in te study very systematically and constructively.
The results of the research are demonstrated visually, purposefully and scientifically. Part of the discussion is described by maintaining links with the singled out goal, problematic issues, hypotheses.
Author Response
Good day!
Dear Reviewer 1,
Thank you for your valuable time and efforts in reading our paper.
Regards,
Klint Allen A. Mariñas
Response to Reviewer 1 Comments
Point 1: In this study „A cognitive Capacity of Modular Distance learning towards academic achievements and satisfaction of K-12 students amidst the COVID-19 Pandemic“ aimed to spot the impact of modular distance learning on the K-12 students amid the COVID-19 pandemic and assess the cognitive factors affecting academic achievement and student satisfaction.
The results of this investigation support the hypothesis that mental traits have the most significant, the biggest, and most important impact on student performance.
Theoretical research framework, materials and methods is described in the study very systematically and constructively.
The results of the research are demonstrated visually, purposefully and scientifically. Part of the discussion is described by maintaining links with the singled out goal, problematic issues, hypotheses.
Response 1: The authors would like to express their sincere gratitude to the Reviewer #1 for his/her valuable comments and for reading our paper.
Reviewer 2 Report
The article is aimed at studying an important and urgent problem. There are a lot of studies of distance education during the covid-19 pandemic. This study compares favorably with the exact theoretical and methodological foundations, which are highlighted in the Abstract and in the Introduction (Transactional Distance Theory and Bloom's Taxonomy Theory). These theories reveal the issues under study well.
However, in my opinion, they are not sufficiently disclosed in sections 1.1 and 1.2 in connection with the current study. These sections can be improved with a more complete justification of the connections of the aim and hypotheses of the study with the main theoretical provisions.
But my main concern is related to the correspondence between the title of the article and the content of the study. According to the title of the article, the subject of the study is cognitive capacities (it is unclear from the title whose they are?). However, cognitive capacities, in my opinion, are not empirically studied. The authors consider various factors affecting academic achievement, academic satisfaction, and perceived effectiveness. Perhaps the title of the article needs to be clarified or the possibility of using the term "cognitive abilities" should be justified in more detail. The last sentence from the Introduction section also needs clarification ("The results support the hypothesis that mental traits have the most significant, the biggest, and most important impact on student performance"). It seems to me that the authors did not study mental traits. The results of the study may not be formulated in the "Introduction", but if the authors consider it necessary, then they should correct this proposal.
Author Response
Good day!
Dear Reviewer 2,
The authors would like to express their gratitude with the reviewer in giving your time and valuable comments and suggestions to the authors in improving the paper.
Thank you and stay safe!
Regards,
Klint Allen A. Mariñas
Response to Reviewer 2 Comments
The authors would like to express their sincere gratitude to the Reviewer #2 for his/her valuable comments and for reading our paper. The response to the comments are as follows.
Point 1: education during the covid-19 pandemic. This study compares favorably with the exact theoretical and methodological foundations, which are highlighted in the Abstract and in the Introduction (Transactional Distance Theory and Bloom's Taxonomy Theory). These theories reveal the issues under study well. However, in my opinion, they are not sufficiently disclosed in sections 1.1 and 1.2 in connection with the current study. These sections can be improved with a more complete justification of the connections of the aim and hypotheses of the study with the main theoretical provisions.
Response 1: The authors expound more and relate it to the aims and hypotheses by adding this statements to improve the connection and justifications of the theories to the aim and hypotheses (page 2)
The TDT was utilized since it has the capability to know the psychological and communication factors between the learners and the instructors in distance education that could eventually help researchers in identifying the variables that might affect students’ academic achievement and satisfaction [67]. In view, distance learning is primarily determined by the number of dialogues between student and teacher and the degree of structuring of the course design. It contributes to the core objective of the degree to boost students' modular learning experiences in terms of satisfaction. On the other hand, Bloom's Taxonomy Theory (BTT) was applied to investigate the students’ academic achievements through the modular distance learning [68]. Bloom's theory was employed in addition to TDT during this study to enhance students' modular educational experiences. In contrast, TDT utilized this study to check students' modular learning experiences in conjunction with their students' achievements to enhance.
- Ekwunife-Orakwue, K. C. V., & Teng, T.-L. (2014). The impact of transactional distance dialogic interactions on student learning outcomes in online and blended environments. Computers & Education, 78, 414–427. https://doi.org/10.1016/j.compedu.2014.06.011
- Abuhassna, H., Al-Rahmi, W. M., Yahya, N., Zakaria, M. A., Kosnin, A. B., & Darwish, M. (2020). Development of a new model on utilizing online learning platforms to improve students’ academic achievements and satisfaction. International Journal of Educational Technology in Higher Education, 17(1). https://doi.org/10.1186/s41239-020-00216-z
Point 2: But my main concern is related to the correspondence between the title of the article and the content of the study. According to the title of the article, the subject of the study is cognitive capacities (it is unclear from the title whose they are?). However, cognitive capacities, in my opinion, are not empirically studied. The authors consider various factors affecting academic achievement, academic satisfaction, and perceived effectiveness. Perhaps the title of the article needs to be clarified or the possibility of using the term "cognitive abilities" should be justified in more detail.
Response 2: Revised the title from A cognitive capacity of Modular Distance learning towards academic achievements and satisfaction of K-12 students amidst the COVID-19 Pandemic to Assessing cognitive factors of Modular Distance learning of K-12 students amidst the COVID-19 Pandemic towards academic achievements and satisfaction. This will avoid confusion from the readers and the goal of the research since the research focused on assessing the cognitive factors.
Point 3: The last sentence from the Introduction section also needs clarification ("The results support the hypothesis that mental traits have the most significant, the biggest, and most important impact on student performance”). It seems to me that the authors did not study mental traits. The results of the study may not be formulated in the "Introduction", but if the authors consider it necessary, then they should correct this proposal.
Response 3: This statements was removed from the manuscript.
Reviewer 3 Report
The article presents a study to relate students' prior background in distance learning during the covid-19 pandemic.
Formally the article meets the expected requirements for this type of research. The theoretical foundations are well chosen and implemented. The model built to analyze the data obtained. Regarding the participants, it would be convenient to comment on the percentage of students responding to the survey with respect to those potentially surveyed, since this data can help to interpret possible response biases.
The quantitative analysis processes are adequate and well justified, so that the results obtained are clear and conclusive. What seems to me to be a relevant shortcoming is the discussion of these results, since it is nonexistent and leaves the article without impact at the time of interpreting the results. I therefore ask the authors to embark on the effort of explaining the implications of their study, since without this subsequent process of discussing the results, the study is not useful beyond its local context.
Author Response
Good day!
Dear Reviewer 3,
The authors would like to express their gratitude with the reviewer in giving your time and valuable comments and suggestions to the authors in improving the paper.
Thank you and stay safe!
Regards,
Klint Allen A. Mariñas
Response to Reviewer 3 Comments
The authors would like to express their sincere gratitude to the Reviewer #3 for his/her valuable comments and for reading our paper. The response to the comments are as follows.
Point 1: The article presents a study to relate students' prior background in distance learning during the covid-19 pandemic. Formally the article meets the expected requirements for this type of research. The theoretical foundations are well chosen and implemented. The model built to analyze the data obtained. Regarding the participants, it would be convenient to comment on the percentage of students responding to the survey with respect to those potentially surveyed, since this data can help to interpret possible response biases.
Response 1: The total number of survey sent and the percetage received was included in the Participants part on Page 6. And based on the total online form received of 84% it is within the percentage that will minimize the response bias according to the cited reference.
2.1. Participants
The principal area under study was San Jose, Occidental Mindoro, although other locations were also accepted. The survey took place between February and March 2022, with the target population of K-12 students in Junior and Senior High Schools from grades 7 to 12, aged 12 to 20, who are now implementing the Modular Approach in their studies amid the COVID-19 Pandemic. A 45-item questionnaire was created and circulated online to collect the information. A total of 300 online surveys was sent out and 252 online form was received a total of 84% response rate [69]. According to several experts, the sample size for Structural Equation Modeling (SEM) should be between 200 and 500 [32].
- Wu, M.-J., Zhao, K., & Fils-Aime, F. (2022). Response rates of online surveys in published research: A Meta-analysis. Computers in Human Behavior Reports, 7, 100206. https://doi.org/10.1016/j.chbr.2022.100206
Point 2: The quantitative analysis processes are adequate and well justified, so that the results obtained are clear and conclusive. What seems to me to be a relevant shortcoming is the discussion of these results, since it is nonexistent and leaves the article without impact at the time of interpreting the results. I therefore ask the authors to embark on the effort of explaining the implications of their study, since without this subsequent process of discussing the results, the study is not useful beyond its local context.
Response 2: The discussion part of the manuscript was improved based on the reviewers comments and suggestions (Page 14)
Regarding the Students' Understanding Response, the results revealed that SAA (β:0.307; p = 0.052) and SS (β:0.699; p = 0.008) had a substantial impact on SAU. Modular teaching is concerned with each student as an individual with their specific capability and interest to assist each K-12 student in learning and provide quality education by allowing individuality to each learner. According to the Department of Education, academic achievement is the new level for student learning [62]. Meanwhile, SAA was significantly affected by the Students' Performance Response (β: 0.754; p = 0.014). It implies that a positive performance can give positive results in the student's academic achievement, and a negative performance can also give negative results {73]. Pekrun et al. (2010) discovered that students' academic emotions are linked to their performance, academic achievement, personality, and classroom circumstances [25].
Results showed that students' academic achievement significantly positively affects SPE (β: 0.237; p = 0.024). Prior knowledge has had an indirect effect on academic ac-complishment. It influences the amount and type of current learning system where students must obtain a high degree of mastery [63]. According to the student's opinion, Modular Distance Learning is an alternative solution for providing adequate education for all learners and at all levels in the current scenario under the new education policy [64]. However, the SEM revealed that SS significantly affected SPE (β:0.868; p = 0.009). Stu-dents' perceptions of learning and Satisfaction, when combined, can provide a better knowledge of learning achievement [65]. Students' perceptions of learning outcomes are an excellent predictor of student satisfaction.
Since p-values and the indicators in Students' Behavior are below 0.5, therefore two paths connecting SBE to students' interaction - instructor (0.155) and students' under-standing (0.212) are not significant; thus, the latent variable Students' Behavior has no effect on the latent variable Students' Satisfaction and academic achievement as well as Perceived Effectiveness on Modular Distance Learning of K12 students. This result is supported by Samsen-Bronsveld et al. (2022), who revealed that the environment has no direct influence on the student's satisfaction, behavior engagement, and motivation to study [70]. On the other hand, the results also showed no significant relationship between Students' Performance and Students Satisfaction (0.602) because the correlation P-values are greater than 0.5. Interestingly, this result opposed the other related studies. According to Bossman & Agyei (2022), satisfaction significantly affects performance or learning outcomes [71]. In addition, it was also discovered that the main drivers of the students' performance are the students' satisfaction [72,73].
The result of the study implies that the students' satisfaction serves as the mediator between the students' performance and the student-instructor interaction in modular distance learning for the K-12 students [74].
- Ekwunife-Orakwue, K. C. V., & Teng, T.-L. (2014). The impact of transactional distance dialogic interactions on student learning outcomes in online and blended environments. Computers & Education, 78, 414–427. https://doi.org/10.1016/j.compedu.2014.06.011
- Abuhassna, H., Al-Rahmi, W. M., Yahya, N., Zakaria, M. A., Kosnin, A. B., & Darwish, M. (2020). Development of a new model on utilizing online learning platforms to improve students’ academic achievements and satisfaction. International Journal of Educational Technology in Higher Education, 17(1). https://doi.org/10.1186/s41239-020-00216-z
- Wu, M.-J., Zhao, K., & Fils-Aime, F. (2022). Response rates of online surveys in published research: A Meta-analysis. Computers in Human Behavior Reports, 7, 100206. https://doi.org/10.1016/j.chbr.2022.100206
- Samsen-Bronsveld, H. E., van der Ven, S. H. G., Bogaerts, S., Greven, C. U., & Bakx, A. W. E. A. (2022). Sensory processing sensitivity does not moderate the relationship between need satisfaction, motivation and behavioral engagement in primary school students. Personality and Individual Differences, 195, 111678. https://doi.org/10.1016/j.paid.2022.111678
- Islam, A. K. M. N., & Azad, N. (2015). Satisfaction and continuance with a learning management system. The International Journal of Information and Learning Technology, 32(2), 109–123. https://doi.org/10.1108/ijilt-09-2014-0020
- Pérez-Pérez, M., Serrano-Bedia, A. M., & García-Piqueres, G. (2019). An analysis of factors affecting students´ perceptions of learning outcomes with Moodle. Journal of Further and Higher Education, 44(8), 1114–1129. https://doi.org/10.1080/0309877x.2019.1664730
- Nja, C. O., Orim, R. E., Neji, H. A., Ukwetang, J. O., Uwe, U. E., & Ideba, M. A. (2022). Students’ attitude and academic achievement in a flipped classroom. Heliyon, 8(1). https://doi.org/10.1016/j.heliyon.2022.e08792
- Bossman, A., & Agyei, S. K. (2022). Technology and instructor dimensions, e-learning satisfaction, and academic performance of distance students in Ghana. Heliyon, 8(4). https://doi.org/10.1016/j.heliyon.2022.e09200
Round 2
Reviewer 3 Report
The authors have made changes to the article and improved it, now it seems to me that it is in a state where it is publishable.